# Diagnostic Performance Evaluation of Multiparametric Magnetic Resonance Imaging in the Detection of Prostate Cancer with Supervised Machine Learning Methods

**DOI:** 10.3390/diagnostics13040806

**Published:** 2023-02-20

**Authors:** Hamide Nematollahi, Masoud Moslehi, Fahimeh Aminolroayaei, Maryam Maleki, Daryoush Shahbazi-Gahrouei

**Affiliations:** Department of Medical Physics, School of Medicine, Isfahan University of Medical Sciences, Isfahan 81746–73461, Iran

**Keywords:** prostate cancer, multiparametric MRI, machine learning, deep learning, accuracy

## Abstract

Prostate cancer is the second leading cause of cancer-related death in men. Its early and correct diagnosis is of particular importance to controlling and preventing the disease from spreading to other tissues. Artificial intelligence and machine learning have effectively detected and graded several cancers, in particular prostate cancer. The purpose of this review is to show the diagnostic performance (accuracy and area under the curve) of supervised machine learning algorithms in detecting prostate cancer using multiparametric MRI. A comparison was made between the performances of different supervised machine-learning methods. This review study was performed on the recent literature sourced from scientific citation websites such as Google Scholar, PubMed, Scopus, and Web of Science up to the end of January 2023. The findings of this review reveal that supervised machine learning techniques have good performance with high accuracy and area under the curve for prostate cancer diagnosis and prediction using multiparametric MR imaging. Among supervised machine learning methods, deep learning, random forest, and logistic regression algorithms appear to have the best performance.

## 1. Introduction

Prostate cancer (PCa) is the most common cancer in men and the second leading cause of cancer-related death in them [1,2,3]. Various methods are used for PCa screening, though these methods are invasive or have low accuracies, such as digital rectal examination, prostate-specific antigen (PSA) tests, and transrectal ultrasound (TRUS)-guided prostate biopsy [4,5,6,7]. New biomarkers, named 8-hydroxy-2-deoxyguanosine (8-OHdG) and 8-iso-prostaglandin F2α (8-IsoF2α), have been reported. Increased levels of these biomarkers indicate prostate cancer in the patient and they are measured through urine tests. Of course, validating these urinary biomarkers in relation to prostate cancer still requires significant research [8]. Meanwhile, prostate MRI plays a crucial role before a biopsy in patients with raised PSA. Multiparametric magnetic resonance imaging (mp-MRI) is a commonly used imaging procedure for diagnosing PCa. Mp-MRI is recognised as the combination of conventional anatomical MRI and at least two functional magnet resonance sequences: diffusion-weighted imaging (DWI), dynamic contrast-enhanced MRI (DCE-MRI), and, optionally, MR spectroscopy (MRS) [9,10]. Various studies have noted that mp-MRI has good accuracy for diagnosing or determining the grade of prostate cancer [11,12]. Of course, it is more challenging to determine the aggressiveness of cancer using MRI than when it is detected by a physician with good reliability. Recently, various studies have used artificial intelligence and MRI images to diagnose or assess the characterization and severity of cancers, including prostate cancer, to reduce human error, increase the speed of diagnosis and classification, and improve overall efficiency and accuracy [13,14,15]. Indeed, artificial intelligence is beneficial in acquiring important clinical information that can help physicians to provide key and critical opinions about clinical prognosis, diagnosing diseases, and treatment outcomes [16,17].

Artificial Intelligence (AI) describes the capability of a computer to model intelligent behaviour, with minimal human intervention, and to reach a certain goal based on provided data. AI has multiple branches. One of these branches is machine learning (ML). ML describes algorithms used to incorporate intelligence into machines by automatically learning from data [16,18]. There are different types of ML. In general, ML types are branched into four groups: Unsupervised learning, Semi-supervised learning, Supervised learning, and Reinforcement learning [19,20,21]. In Supervised learning, an observer provides data to the machine and labels the data types. Input and output are specified and the machine attempts to learn a pattern from the input to the expected output [22,23]. In unsupervised learning, the computer finds connections between data and discovers patterns without the help of a trainer and without the use of labels that define the type of data [24,25]. Semi-supervised learning is a learning paradigm that studies how computers learn in the attendance of labeled and unlabeled data. During semi-supervised learning, the aim is to design algorithms using combinations of labeled and unlabeled data [26]. Reinforcement learning is conducted by encouraging desirable behaviour and punishing undesirable behaviour. In this way, the computer can understand and interpret various issues by trial and error, according to the feedback it receives as a result of its actions [27,28].

The most common categories of ML algorithms are classification and regression. Examples of supervised learning algorithms include linear and logistic regression, support vector machines (SVMs; classification), K nearest neighbours (KNN; classification and regression), naive Bayes (classification), decision tree and random forests (DT and RF, respectively; both classification), and deep learning techniques (classification) [16,25].

The goal of this review study is to show the diagnostic performance (accuracy and area under the curve) of mp-MR images for predicting prostate cancer with and without using supervised ML learning algorithms. In this review, for a better comparison of the method’s results, studies have been used whose input data included T_1_-weighted imaging (T_1_WI) or T_2_-weighted imaging (T_2_WI), DWI, DCE-MRI, and, optionally, MRS.

### 1.1. mp-MRI in the Detection PCa

Mp-MRI primarily contains at least three sequences: T_2_WI or T_1_WI, DWI, and DCE imaging [29]. T_1_WI is used to detect bleeding after a biopsy. T_2_-weighted images can detect the anatomical shape of the peripheral and transitional zones, where 70% and 30% of cancers are found, respectively [9]. DWI measures the Brownian movement of free water protons inside a tissue. Malignant tissue is denser than normal tissue, triggering restricted free water movement inside the cancerous tissue, thereby decreasing the diffusion of water [30,31,32]. DCE assesses the perfusion and vascular permeability throughout the prostate and within a cancerous tissue through the rapid administration of gadolinium chelates and the use of fast T_1_-weighted images. Unlike normal tissue, malignant tissue has more penetrable, heterogeneous, and disordered vessels due to neoangiogenesis [9,33].

Various studies have used mp-MRI to diagnose PCa and have noted its diagnostic performance. Two examples of mp-MRI diagnostic performance are shown in Figure 1 and Figure 2 [34]. Di Campli et al. [35] conducted a study on mp-MRI to determine the diagnostic accuracy of PCa. A total of 85 patients underwent prostate MRI investigation at a 1.5 T MR system without an endorectal coil. In this study, the MR images were separately interpreted by three radiologists with 7 (reader 1), 3 (reader 2) and 1 year(s) (reader 3) of experience in prostate MRI, respectively (according to Prostate Imaging Reporting and Data System (PI-RADS) version 2). The sensitivity (CI 95%), specificity (CI 95%), area under the curve (AUC), and accuracy values for readers 1, 2, 3 were obtained (97.2% (90.3–99.7%), 88.9% (79.3–95.1%), 83.3% (72.7–91.1%)), (61.5% (31.6–86.1%), 23.1% (5–53.8%), 46.2% (19.2–74.9%)), (0.72, 0.70, 0.54), and 90.58, 78.82, and 77.64, respectively [35].

Kam et al. [36] assessed the accuracy of mpMRI to predict PCa pathology. In their work, 235 patients underwent mpMRI with a 1.5 T or 3 T MRI. The results of mpMRI were compared with the final radical prostatectomy specimen to analyze the performance of mpMRI for significant prostate cancer (sPCa) detection. They reported the accuracy of mpMRI for the prediction of sPCa. Overall, the sensitivity, specificity, and positive predictive value (PPV) of mpMRI for the detection of sPCa were 91%, 23%, and 95%, respectively. In 2020, Ippolito et al. [37] stated the multiparametric diagnostic accuracy of 201 patients for PCa detection. Patients underwent mp-MRI examination with a 3 T MR scanner and a body coil with sequences T_2_WI, DWI, and DCE. The sensitivity, specificity, and accuracy of PI-RADS for the detection of PCa were 65.1%, 54.9%, and 64.2% (55.1–72.7%), respectively.

Consequently, in a study of systematic review and meta-analysis, Zhao et al. [38] reported the diagnostic performance of mp-MRI. The meta-analysis included 10 articles. At a per-patient level, the pooled sensitivity, specificity, and AUC values for mpMRI were 0.87 (0.83–0.91), 0.47 (0.23–0.71), and 0.84, respectively. At a per-lesion level, the pooled sensitivity, specificity, and AUC values were 0.63 (0.52–0.74), 0.88 (0.81–0.95), and 0.83, respectively.

### 1.2. Machine Learning (ML)

ML includes unsupervised, semi-supervised, supervised, and reinforcement learning. In this study, the emphasis is placed on supervised methods that can be employed on data that have been class-labeled for imaging data. There are three primary applications to use ML in medical imaging for tumor diagnosis: localization, segmentation, and classification [16]. The use of a classification model usually includes three stages: training, validation, and testing.

Figure 3 shows the flow diagram of a computer-aided diagnosis system, that begins with MRI procurement and finishes with ML analysis [39].

The purpose is first to develop a computer-aided diagnosis system based on regions of interest (ROIs) drawn by the radiologist. The radiologist then questions the system about a suspicious area and the system returns a probability estimation of malignancy as a reply. For most computer-aided diagnosis systems, this approach can be partitioned into five fundamental steps: MRI acquisition, image segmentation, image processing (resampling, normalization, and discretization), feature extraction (extract multiple parameters of structural, statistical, and functional), and classifier construction and evaluation (classifiers include linear and logistic regression, SVM, KNN, naive Bayes, ANN, DT, and RF) [40].

In this study, supervised machine learning algorithms were evaluated to compare their performances with accuracy, ROC-AUC on prostate cancer diagnostic in classifying cancer and normal tissues, and cancer grading.

The definition and diagnostic performance of supervised machine learning algorithms in prostate cancer detection and prediction in the study are provided.

#### 1.2.1. Detecting/Predicting PCa with mp-MRI Using Linear/Logistic Regression

The function of linear regression is to create a linear relationship to show the relationship between a numeric dependent variable and one or more independent variables. In logistic regression, to specify the model of the relationship between the dependent and independent variable instead of a linear relationship, a “Logistic Function” is used that varies from 0 to 1. This technique uses for data classification. The main feature that distinguishes logistic regression from linear regression is that the dependent variable has two or more classes [16].

Iyama et al. developed a logistic regression model to differentiate transition zone (TZ) cancers and benign prostatic hyperplasia (BPH) on mp-MRI. In this study, 60 patients with BPH or TZ cancer were enrolled. Patients underwent a 3 T MR scanner, a surface coil, and a radical prostatectomy. They calculated the AUC of logistic regression models with a leave-one-out cross-validation procedure [41]. In 2019, Kan et al., in a study of 346 patients with PI-RADS 3 lesions at two institutions, retrospectively collected their data and showed that external validation was performed using a hospital dataset. Patients experienced prostate mpMRI with a 3 T scanner and a surface coil. Two radiologists, using PI-RADS v2.1 standards, managed the results of the images. All lesions of PI-RADS 3 were approved by another radiologist. Finally, all patients experienced both targeted and systematic biopsies to correctly classify the lesions seen on the MRI report. Subsequently, they reported the diagnostic performance of the logistic regression classifier [42].

Alam et al. reported PCa diagnostic and prediction sensitivity and specificity mp-MRI using a modified logistic regression. A total of 387 patients (193 PCa patients and 194 who did not have PCa) were enrolled. Accuracy values were obtained for logistic regression to distinguish cancerous tissue from normal tissue and for cancer prediction [43]. Tang et al. investigated the value of logistic regression combined with mp-MRI in detecting PCa. This study included 64 cases of PCa confirmed by biopsy. After PCa diagnosis, patients underwent radical prostatectomy. Cross-validation of the model was conducted on the external or newly-arrived data [44].

#### 1.2.2. Detecting PCa with mp-MRI Using Support Vector Machines

One of the supervised learning methods is Support Vector Machines (SVM), used for regression and classification. The working basis of the SVM classifier is the linear classification of the data. In the linear division of the data, a line is chosen that has a higher margin of confidence. Assuming that the categories are linearly separable, it obtains hyperplanes with the maximum margin that separates the categories. In cases where the data are not linearly separable, the data are mapped to a space with larger dimensions using a suitable kernel function so that they can be linearly separated in this new space [45].

Niaf et al. [40] have evaluated the performance of SVM for detecting PCa in the peripheral zone (PZ) based on mp-MRI. This study included a series of 42 cancer ROIs, 49 suspicious benign ROIs, and 124 nonsuspicious benign ROIs. Radical prostatectomy was used as the gold standard. The classifier’s performance was assessed using a cross-validation method. The quantitative evaluation of the diagnostic performance of the SVM classifier was obtained for the differentiation of cancerous versus noncancerous tissues and the differentiation of cancerous versus suspicious tissues. Tang et al. [44] retrospectively evaluated 64 patients who underwent mp-MRI before radical prostatectomy to assess the value of AL-combined mp-MRI in PCa detection. SVM-mpMRI achieved a detection rate with an accuracy of 74.9% and an AUC of 0.82.

A recently reported study by Gravina et al. [46] evaluated ML procedures in the diagnosis of PCa in 109 patients with PI-RADS score 3 lesions by focusing on clinical-radiological features. Patients received mp-MRI and transrectal prostate biopsy. The SVM algorithm used a linear kernel. The 10-fold Cross-validation was conducted to evaluate the classifier’s performance.

#### 1.2.3. Detecting PCa with mp-MRI Using k-Nearest Neighbors

The k-nearest neighbor (KNN) algorithm can be used for regression and classification problems. However, it is often used for classification problems. The KNN algorithm requires training data and a specified K value to search for k-nearest data using distance computations. In the classification mode, the algorithm calculates the distance of the point that needs to be labeled with the closest points according to the specified value of K. Then, the label of the desired point is determined according to the maximum number of votes of these neighboring points. Different procedures can be used to compute this distance. One of the most well-known methods is the Euclidian distance. In the case of regression, the average of the values accessed from the K is its output [47].

Anderson et al. trained and tested three diagnostic models: a logistic regression model, a KNN classification algorithm, and a combination of the two. The input data generated from the multiparametric images of PCa patients included apparent diffusion coefficient (ADC), volume transfer constant (Ktrans), a conventional average of T_2_ values, and MRS score. They used leave-one-out cross-validation to separate the data set into a training set and a test set. Finally, they investigated the performance of three models in detecting and classifying the degree of malignancy of PCa [48].

Various studies have used the KNN algorithm to diagnose prostate cancer using multiparametric images and noted its diagnostic performance. A study reported the diagnostic performance of the KNN with an accuracy of 78.75% [43]. Another study reported an AUC value of 0.88 (0.81–0.92) for the differentiation of cancerous from noncancerous tissues when combining the *t*-test property selection procedure with a KNN classifier [40].

#### 1.2.4. Detecting PCa with mp-MRI Using Decision Tree/Random Forest

Decision tree (DT) and random forest (RF) can be used for both classification and regression problems. One of the advantages of a DT is that we can draw the entire trained model, drawn upside down with its root at the top. The DT is one of the most interpretable models in ML. It uses a series of decision rules. It expands with the first decision rule at the top (the root of the tree) and subsequent decision rules below, which are called nodes. In a DT, a decision rule occurs at each decision node, that then leads to new nodes. At the end of each tree, it reaches the ‘leaf’, which is the goal of the problem and determines the class. Meanwhile, RF creates a forest randomly. The built “forest” is a group of “decision trees”. The work of making a forest using trees is often performed by the “bagging” method. The principal idea of the bagging method is that the combination of learning models increases the overall results of the model. Simply put, an RF builds multiple DTs and merges them to produce more accurate and stable predictions [49,50].

In 2021, Peng et al. [51] developed ML models including a DT and RF for the diagnosis of PCa with a Gleason score ≥7 using mp-MRI, texture analysis, DCE-MRI quantitative analysis, and clinical parameters. In their study, a dataset of 194 patients was collected and the mp-MRI using a 1.5 T system with a combined spine and body-array coil was performed before the target biopsy. The ROIs in temporal validation were independently delineated by two radiologists (Doctor A and Doctor B with 10 and 3 years of experience in prostate MRI, respectively.) they then stated the accuracy values for Doctor A and Doctor B with DT and RF models [51].

In some studies, the values of the diagnostic performance of DT and RF using mp-MRI in PCa patients have shown the sensitivity, specificity, overall accuracy, and AUC under the default threshold for the RF classifier by lesion to be 0.613, 0.952, 0.860, and 0.832, respectively [42]; the accuracy of the DT and RF classifier to be 77.95% and 92.84%, respectively [43]; and the accuracy and AUC of the RF classifier in 10-fold cross-validation to be 77.98% and 83.32%, respectively [46].

#### 1.2.5. Detection PCa with mp-MRI Using Naive Bayes

The Naïve Bayes algorithm is a classification method based on the application of the Bayes theorem with the strong assumption that all predictors are independent of each other. This method is considered one of the simplest forecasting algorithms and has a very acceptable accuracy, which are both its advantages [52].

Alfano et al. developed a radiomics-based Naïve Bayes algorithm to detect cancerous tissue from noncancerous tissue of the prostate using mp-MRI. They reported that the AUC value for a 5-feature Naïve-Bayes classifier was 0.80 and this was validated using leave-one-patient-out cross-validation. To detect differences in shape between cancerous tissue and noncancerous tissue of the prostate, classifiers were invariant (AUC: 0.82). Performance for models trained and tested in the PZ (AUC: 0.75) was lower than in the central gland (AUC 0.95) [53]. In the study mentioned in the previous sections, Niaf et al. obtained the Naïve-Bayes classifier AUC value of 0.88 [0.82–0.93] and 0.77 [0.66–0.85] for the differentiation of cancerous from noncancerous tissues and the differentiation of cancerous from suspicious tissues, respectively [40].

#### 1.2.6. Detection PCa with mp-MRI Using Artificial Neural Network

Artificial neural network (ANN) training can be divided into unsupervised learning, supervised learning, and rain forest learning [24]. ANN is one of the primary ML methods used for data classification. The central premise of ANN is inspired by the way the biological nervous system works, such as the way the brain processes data and information to learn and create knowledge. An ANN consists of three layers: input, data transforming or hidden, and output. Each layer contains a group of connected computational units (neurons). The network is trained to create accurate predictions by recognizing predictive properties in a set of labeled training data, while the outputs are compared with the true labels by an objective function [16]. Deep learning (DL) or deep neural networks (DNNs) are the more complex form of ANN with multi-layer perceptrons. DNN algorithms require a significant amount of data and equipment with exceptionally tall computing control that can handle this data. In this algorithm, features are automatically and directly derived from the crude imaging data and optimally adjusted for the desired result. In medical imaging, DL is often produced by convolutional neural networks (CNNs) [16,54].

Kiraly et al. [55] have proposed a deep convolutional neural network (DCNN) for PCa detection and classification. The conclusive diagnosis was determined by histopathological examination of tissue biopsies. The input data consisted of T_2_W, ADC, high b-value DWI, and Ktrans parameter maps (images were obtained using DCE) from the ProstateX challenge database for 202 patients. The value of AUC was obtained with 5-fold cross-validation across. In 2018, Wang et al. developed a fully convolutional neural network using mp-MRI including T_2_W, DWI, ADC, and K-trans for PCa detection. The dataset used for the development of tumor detection contained volumetric prostate MR images acquired from 79 patients. The CNN generated an accuracy level of 0.85 [56]. In 2022, Pellicer Valero et al. [57] employed an automatic system based on DL that performed localization, segmentation, and Gleason grade group (GGG) prediction of PCa in mp-MRIs. The prostate mp-MRI datasets of the Valencian Oncology Institute Foundation (IVO) and ProstateX, which is part of an ongoing online challenge, were used for the development and validation of the model. Using ProstateX and IVO, the data included a total of 204 and 221 mp-MRIs, respectively. The physician’s report of the lesion’s locations were confirmed by MR-guided biopsy. Data were divided into four PCa categories: GGG0 or benign, GGG1 (Gleason Score (GS) 3 + 3), GGG2 (GS 3 + 4), and GGG3+ (GS ≥ 4 + 3). In the test dataset, at a lesion and patient level using the ProstateX and the IVO dataset, the DL achieved good diagnostic performance.

## 2. Discussion

In this review study, we showed the diagnostic performance of mp-MRI images for diagnosing or predicting PCa with and without using ML-supervised learning algorithms. In recent years, many studies have been published, showing the use of ML methods on mp-MRI to detect and classify PCa to compare the diagnostic performance of ML methods and radiologist, to provide intelligent methods to increase the speed of diagnosis and classification, reduce human error and prevent from unnecessary biopsies, and to help the radiologist in their diagnostic workflow. Diagnosis of PCa and its grade is of vital to control, treat, and prevent the disease from spreading to other tissues. Successful treatment of this cancer requires its early diagnosis, and for this purpose, pathological examination of the tissue sample is necessary [58]. According to studies, 70% of PCa occurs in the PZ and 30% in the TZ. The TZ is also the place of BPH, a non-cancerous growth, that can result in urinary obstruction. BPH appears to mimic PCa on mp-MRI, thus rendering the diagnosis of PCa in the TZ difficult. This triggers unnecessary biopsies and identifies various undiagnosed cancerous lesions that indicate progression [1]. According to studies, PSA value, biological, and mp-MRI information without machine learning have low specificity (significant overdiagnosis) in PCa diagnosis, leading to unnecessary biopsies and to significant infectious complications, psychological harm, and financial costs [59,60,61]. A radiologist’s diagnostic performance is influenced by their skill level and, perhaps, experience (not definitively), and possibly the PI-RADS category version.

In a study using three radiologists with 7, 3, and 1 years of experience in diagnosing PCa, Campelli et al. reported that there were no significant differences between the ROC curves for each protocol between the most experienced radiologist and the others. [35].

Kam et al. compared mp-MRI cases using the technical and reporting specifications of PI-RADS version 1 and version 2. The sensitivity for prediction of significant PCa was lower in the PI-RADS version 1 (87%) cases when compared with PI-RADS version 2 (99%, *p* = 0.005) [36].

Therefore, due to the possibility of affecting the diagnostic performance of radiologists, the reduction of unnecessary biopsies, and reducing the time spent on recognition and diagnosis, it is better to make use of AI with good performance. As noted above, AI describes the capability of a computer to model intelligent behavior and to reach a specific goal based on provided data. One of the branches of AI is ML. In supervised machine learning methods, an observer provides data to the machine and labels the data types. In this type of learning, the input and output are specified, and the machine tries to learn a pattern from the input to the expected output. Examples of supervised learning algorithms whose diagnostic performance was examined in this review study include linear and logistic regression, K nearest neighbors (KNN), support vector machines (SVMs), naive Bayes, decision tree (DT) and random forests (RF), and artificial Neural Network (ANN) techniques. Table 1 shows the diagnostic performance (accuracy or AUC) of mp-MRI in the detection of PCa with supervised machine learning algorithms noted in several studies. Table 2 shows the results of several works that have stated the algorithms with better diagnostic performance among the studied algorithms.

In a study, Kan et al. [42] used prostate mp-MRI and clinical-radiological features-based ML to state if a PI-RADS 3 patient was benign, using logistic regression (LR), SVM, and RF classifiers. They reported that the RF classifier had the best performance in both lesion-based and patient-based datasets, with a sensitivity of 0.613 and 0.857, an overall accuracy of 0.860 and 0.713, and an AUC of 0.832 and 0.771, respectively. In another study, Gravina et al. [46] compared the algorithms’ performance in RF, SVM, and neural networks in the diagnosis of PCa in patients with PI-RADS score 3 lesions with attention to clinical-radiological features. The patients underwent mp-MRI. In total, they reported the RF had the best performance, with an AUC of 83.32%. The value of the AUC of the NN and SVM were reported at 74.51% and 72.76%, respectively. Tao Peng et al., established ML models including LR, classical decision tree (cDT), RF, and SVM for the diagnosis of clinically significant prostate cancer (csPC) using mp-MRI and clinical parameters. They reported that the RF and LR models had better classification performance in the imaging-based diagnosis of csPC [51]. Donisi et al. combined radionics and ML approaches (RF, NB, and KNN) to distinguish csPC lesions on T_2_W and ADC maps images. They concluded the best algorithm is RF, due to its high accuracy (77.9%) and AUC (0.73). However, NB obtained the highest sensitivity (56.6%) while KNN had the highest specificity (91.9%) [39]. Chiu et al. [62] compared the values of AUC, of PSA, PSA density, and techniques of logistic regression, SVM, and RF using PSA, digital rectal examination (DRE), and transrectal ultrasound (TRUS) prostate volume information in the prediction of any grade PCa. Their results revealed that, in PCa prediction, all ML techniques achieved better AUC than PSA (AUC 0.67, 95% CI 0.63–0.71) or PSA density (AUC 0.75, 95% CI 0.71–0.80). ML techniques using the same clinical parameters can improve the PCa prediction when compared with PSA and PSA density and prevent up to 50% unnecessary biopsies. The RF model achieved the best AUC among other ML models (AUC 0.82, 95% CI 0.78–0.86).

Therefore, according to the studies in the literature, the RF model is likely the best ML model compared to LR, SVM, NB, NN, KNN, and cDT. The reasons for this superiority can be found in the RF model’s features and advantages compared with other ML models. RF is less susceptible to overfitting and decreases overfitting in DTs, helping to improve accuracy. The output is the most important feature, which is very useful for model analysis. It has good performance with categorical and continuous values for classification and regression problems. It solves lost values in the data by automating the analysis of lost values. The output changes significantly with minor changes within the data. Meanwhile, SVM and KNN are prone to overfitting and sensitive to noise. In addition, KNN is sensitive to missing data and SVM is sensitive to large datasets. NB has difficulties with complex datasets as they are linear classifiers. Logistic regression assumes linearity between dependent and independent variables, and linear relationships between variables are rare [16]. Despite this, in a study on the prediction of PCa, it was shown that RF and LR both have better accuracy (90%) when compared to other ML algorithms, including SVM, KNN, and NB [63]. In another previously mentioned study, it was reported that RF and LR perform better than SVM and cDT in the diagnosis of csPC using mp-MRI [51].

Currently, deep neural networks (DNN) are the most advanced ML models in various domains, outperforming other established modeling methods in several important metrics [16]. Wang et al. compared the performance of DL with a DCNN algorithm and a non-DL (SVM) method to differentiate cancerous from non-cancerous prostate tissue using images of T_2_WI, T_1_WI, DCE-MRI, and DWI. The values of AUC, sensitivity, and specificity for DL and non-DL methods were achieved at 0.84, 69.6%, and 83.9% and 0.70, 49.4%, and 81.7%, respectively [56]. In a non-medical study, a comparison was made between the performance of DL (DNN and CNN) and RF methods for predicting monthly evaporation pan rates. The results revealed that higher accuracy was obtained with the DL models (DNN and CNN) for predicting evaporation versus a RF model. This can be associated with the DL characteristic of revealing hidden features, meaning that DL can be regarded as a better method for prediction [64]. Tabares-Soto et al. compared the performance of ML and DL algorithms for the classification of cancer species based on microarray gene expression data. They obtained the highest tumor identification accuracies for CNN (94.43%) and for LR (90.6%) using 10-fold cross-validation [65]. Non-deep-learning methods such as KNN, BNs, SVM, DTs, and ANNs depend on a feature extraction step that typically describes images using texture, gradient, and Gabor filters, etc. Despite this, the superior performance of the deep learning method is because it learns image features automatically in deep networks.

In recent years, high-frequency ultrasound imaging (micro-ultrasound) has been introduced in the urological field and many advances have been made as a result. Various studies have shown that high-resolution micro-ultrasound has the same or even higher prostate cancer detection ability than mp-MRI [66,67]. Compared to the mp-MRI method, this method has advantages such as easy access, simplicity, and lower cost [67]. Our suggestion is to produce a review article showing the diagnostic performance of micro-ultrasound in the detection of prostate cancer with supervised or unsupervised machine learning methods and compare it with mp-MRI.

## 3. Conclusions

PSA value, biological, and mp-MRI information without machine learning has low specificity (significant over-diagnosis in prostate cancer diagnosis, leading to unnecessary biopsy and side effects such as infectious complications, psychological harm, and financial costs). But, when this information is given as input data to supervised machine learning algorithms, it increases the sensitivity, specificity, and the accuracy of prostate cancer diagnosis and prediction. It appears that deep learning, random forest, and logistic regression algorithms have the best performance among supervised machine learning techniques.

## Figures and Tables

**Figure 1 diagnostics-13-00806-f001:**
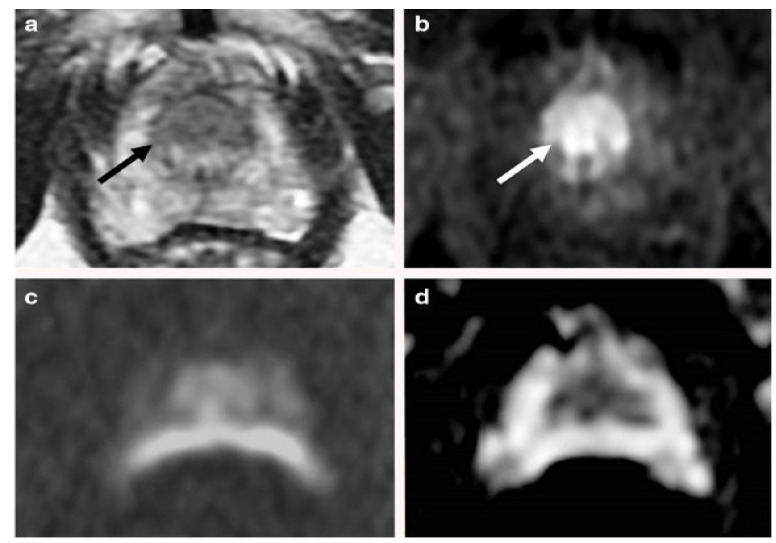
An example of mp-MRI diagnostic performance in a 66-year-old man with PSA 9.1 ng/mL. A focal low signal (arrow) in the midline apex TZ, (**a**) but there is not a high signal on the DWI, (**c**) or low signal on ADC maps (**d**). An early and clear enhancement on the DCE-MRI (arrow) in the midline apex TZ (**b**) was recognized as a high-possibility lesion. This lesion was proven by targeted transperineal biopsy (Gleason 5 + 4). “Reprinted with permission from Ref. [34]. 2020, Springer”. More details on “Copyright and Licensing” are available via the following link: https://link.springer.com/article/10.1007/s00330-020-06782-0 (accessed on 12 March 2020).

**Figure 2 diagnostics-13-00806-f002:**
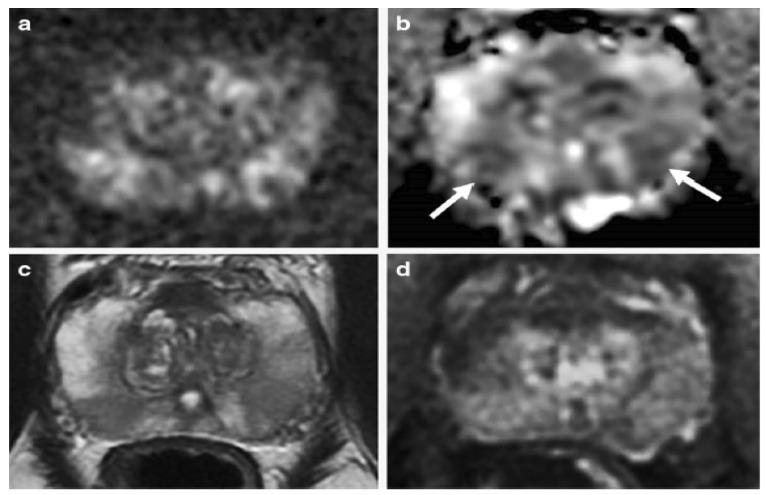
An example of mp-MRI diagnostic performance in a 62-year-old man with PSA 6.04 ng/mL. The DWI (**a**) and ADC maps (arrows) (**b**) showed a mild restricted diffusion in the bilateral base PZ. Unclear signal intensity on the T_2_WI (**c**) and a diffuse wedge-shaped enhancement on the DCE-MRI (**d**), imagined showing an inflammatory change. A systematic TRUS biopsy was performed with negative cores. “Reprinted with permission from Ref. [34]. 2020, Springer”. More details on “Copyright and Licensing” are available via the following link: https://link.springer.com/article/10.1007/s00330-020-06782-0 (accessed on 12 March 2020).

**Figure 3 diagnostics-13-00806-f003:**
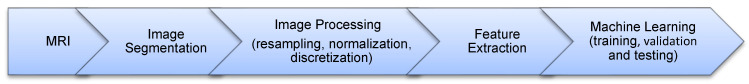
General data flow diagram of a computer-aided diagnosis system.

**Table 1 diagnostics-13-00806-t001:** The diagnostic performance (accuracy or AUC) of mp-MRI in the detection of PCa with supervised machine learning algorithms was mentioned in several studies.

Author	Studied Algorithm	The Value of Accuracy	The Value of AUC
Iyama et al. [41]	LR	-	0.97
Kan et al. [42]	LR	0.862	0.735
Alam et al. [43]	LR	0.969	-
Tang et al. [44]	LR	0.754	0.82
Niaf et al. [40]	SVM	-	0.89
Tang et al. [44]	SVM	0.749	0.82
Gravina et al. [46]	SVM	0.725	0.727
Anderson et al. [48]	KNN	0.77	0.82
Alam et al. [43]	KNN	0.787	-
Niaf et al. [40]	KNN	-	0.88
Kan et al. [42]	RF	0.860	0.832
Alam et al. [43]	DT	0.779	-
Alam et al. [43]	RF	0.928	-
Gravina et al. [46]	RF	0.779	0.833
Alfano et al. [53]	NB	-	0.80
Niaf et al. [40]	NB	-	0.88
Kiraly et al. [55]	DCNN	-	0.83
Wang et al. [56]	CNN	0.85	-

Abbreviations; LR: logistic regression, SVM: support vector machines, RF: random forests, cDT: decision tree, KNN: K nearest neighbors, NB: naive Bayes, DNN: deep neural network, DL: Deep learning, CNN: convolutional neural network.

**Table 2 diagnostics-13-00806-t002:** Algorithms with the better diagnostic performance among the studied algorithms in the literature.

Author	Studied Algorithms	The Best Algorithms
Kan et al. [42]	LR, SVM, and RF	RF
Gravina et al. [46]	RF, SVM, and neural network	RF
Peng et al. [51]	LR, cDT, RF, and SVM	RF and LR
Donisi et al. [39]	RF, NB, and KNN	RF
Ka-Fung Chiu et al. [62]	LR, SVM and RF	RF
Srivenkatesh. [63]	SVM, KNN and NB	RF and LR
Wang et al. [57]	DCNN and SVM	DCNN
Abed et al. [64]	DL (DNN and CNN) and RF methods	DL
Tabares-Soto et al. [65]	ML and DL algorithms	CNN and LR

Abbreviations; LR: logistic regression, SVM: support vector machines, RF: random forests, cDT: decision tree, KNN: K nearest neighbors, NB: naive Bayes, DNN: deep neural network, DL: Deep learning, CNN: convolutional neural network.

## Data Availability

The data presented in this study are available on request from the corresponding author.

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
