# Peer review of "Diagnostic Performance Evaluation of Multiparametric Magnetic Resonance Imaging in the Detection of Prostate Cancer with Supervised Machine Learning Methods"

_diagnostics, 2023, doi:10.3390/diagnostics13040806_

Round 1
Reviewer 1 Report
The paper is well-written, good structured and very informative. It emphasizes the role of multiparametric MRI in the diagnosis of prostate cancer using supervised machine learning methods simply and clearly.
Author Response
Thanks for your taking the time to consideration my manuscript.important and valuable comments.

Reviewer 2 Report
The abstract needs quantification. section needs to discussion about the need for this research. Section 2 will be review only. Results are not fully explored. The discussion part is very poor. Cost analysis may be provided. F1 score, MCC and Kappa may be included. Dice and Jaccard score may also be included. Table 1 needs parameter comparison. The reference are adequate. Figure 1 is general needs modification with specific functions of the article.
Author Response
Thanks for important and valuable comments. All changes were done accordingly and highlighted in the revised file with yellow color.
- English language and style are fine/minor spell check required
English language and style were corrected.
2.1. The abstract needs quantification.
This study is a narrative review article that cannot express the final value of the desired result. In fact, this study is a qualitative study. If it was a meta-analysis review article, the value would have been mentioned. Hence, we just mentioned at the end that among supervised machine learning methods, it seems that deep learning, random forest, and logistic regression algorithms have the best performance.
2.2. section needs to discussion about the need for this research. Section 2 will be review only.
It was modified as indicated in this section.
This sentence was added:
The purpose of this study is to show the diagnostic performance (accuracy and area under curve) supervised learning machines in the detection of prostate cancer on multiparametric MRI.
- Results are not fully explored.
The goal of this review study is to show the diagnostic performance (accuracy or area under the curve) of mp-MR images for predicting prostate cancer with and without using ML-supervised learning algorithms. And we tried to bring the articles that stated the accuracy or the area under the curve of supervised machines learning methods using multiparametric MR images.
- The discussion part is very poor. Cost analysis may be provided. F1 score, MCC and Kappa may be included.
Dice and Jaccard score may also be included.
Major modification was performed in the discussion section. Since the purpose of this study is to show the diagnostic performance (accuracy and area under curve) supervised learning machines in the detection of prostate cancer, for this reason only studies that reported the accuracy and area under the curve of supervised learning machines are included.
- Table 1 needs parameter comparison. The reference are adequate.
Explanations of abbreviations were added in the table footnote.
- Figure 1 is general needs modification with specific functions of the article.
Figures 1-2 are presented in the section of mp-MRI in the detection Pca.

Reviewer 3 Report
This review aims to show the diagnostic performance of mp-MR images for predicting prostate cancer with and without using ML-supervised learning algorithms. The manuscript needs deep changes and improvements.
- the introduction should be reworded. Please restructure the introduction by shortening and eliminating data and comments from other studies that even if valuable should be included in the discussion. Indeed, some concepts are retracted in the discussion, making the whole text redundant and difficult to read.
- Figures 1-2 should be presented in a different section (not in the introduction)
- when introducing an abbreviation please be consistent with its use. (e-g for example Artificial intelligence)
- Review study = review or narrative review
- Why did you present a result section? you did not provide any results. Your paper is a review. eliminate it.
- Because you presented an overview of all available tests for PCa. Please include a novel finding. Indeed, a novel urinary biomarker has been described (8-OHdG and 8-Iso-PGF2α) in the urine of patients with prostate cancer. The measurement of 8-OHdG and of 8-Iso-PGF2α seems to be related to cancer radicality (and perhaps local recurrence) following surgery (DOI: 10.3390/jcm11206102). I strongly believe this should be included in your paper.
- Novel radiological modalities have also been proposed for PCa in recent years. Micro-ultrasound (MUS) is an imaging examination characterized by high real-time spatial resolution, recently introduced in the urological field
In the discussion, please include the following recent paper on the topic (10.3390/medicina58111624).
- please shorten the conclusion, eliminate the brief redundant introduction
- check typos
Author Response
Thanks for your important and valuable comments. All changes were done accordingly and highlighted in the revised file with yellow color.
Comments and Suggestions for Authors
This review aims to show the diagnostic performance of mp-MR images for predicting prostate cancer with and without using ML-supervised learning algorithms. The manuscript needs deep changes and improvements.
1- the introduction should be reworded. Please restructure the introduction by shortening and eliminating data and comments from other studies that even if valuable should be included in the discussion. Indeed, some concepts are retracted in the discussion, making the whole text redundant and difficult to read.
Major modification was performed in the introduction and discussion sections.
2- Figures 1-2 should be presented in a different section (not in the introduction)
Figures 1-2 are presented in the section of mp-MRI in the detection Pca.
3- when introducing an abbreviation please be consistent with its use. (e-g for example Artificial intelligence)
All of them were corrected.
4- Review study = review or narrative review
It is a narrative review
5- Why did you present a result section? you did not provide any results. Your paper is a review. eliminate it.
Thanks for your helpful comment. This subtitle was deleted.
6- Because you presented an overview of all available tests for PCa. Please include a novel finding. Indeed, a novel urinary biomarker has been described (8-OHdG and 8-Iso-PGF2α) in the urine of patients with prostate cancer. The measurement of 8-OHdG and of 8-Iso-PGF2α seems to be related to cancer radicality (and perhaps local recurrence) following surgery (DOI: 10.3390/jcm11206102). I strongly believe this should be included in your paper.
Thanks for your valuable comment. It was added in the Introduction section as follow:
New biomarkers named 8-hydroxy-2-deoxyguanosine (8-OHdG) and 8-iso-prostaglandin F2α (8-IsoF2α) have been reported, whose level increase indicates prostate cancer in the patient, which is measured through urine tests. Of course, validating these urinary biomarkers in relation to prostate cancer still needs a lot of research (8).
7- Novel radiological modalities have also been proposed for PCa in recent years. Micro-ultrasound (MUS) is an imaging examination characterized by high real-time spatial resolution, recently introduced in the urological field
In the discussion, please include the following recent paper on the topic (10.3390/medicina58111624).
Thanks for your valuable comment. I did modify and added it as follow:
In recent years, in the urological field, high-frequency ultrasound imaging (micro-ultrasound) is introduced and many advances have been made in this field. Various studies have shown that high-resolution micro-ultrasound has the same or even higher prostate cancer detection ability than mp-MRI (68, 69). Compared to the mp-MRI method, this method has advantages such as easy access, simplicity and lower cost (69). Our suggestion is to do a review article to show the diagnostic performance of micro-ultrasound in the detection of prostate cancer with supervised or unsupervised machine learning methods and compare it with mp-MRI.
9- please shorten the conclusion, eliminate the brief redundant introduction
It was modified accordingly.
10- check typos
The Typo and English language were checked.

Round 2
Reviewer 2 Report
All the corrections are included and the paper may be accepted.
Reviewer 3 Report
The revised manuscript has been deeply improved. In my opinion, is worthy of publication